# Evacuation and Transportation Barriers Among Vulnerable Populations in Natural Hazard-Related Disasters: A Scoping Review

**DOI:** 10.3390/ijerph22111680

**Published:** 2025-11-05

**Authors:** Yuriko Matsuo, Kathryn Kietzman, Ron D. Hays, Yeonsu Song

**Affiliations:** 1Joe C. Wen School of Nursing, University of California, Los Angeles, CA 90095, USA; ymatsuo@sonnet.ucla.edu; 2Center for Health Policy Research, University of California, Los Angeles, CA 90024, USA; 3Department of Community Health Sciences, Fielding School of Public Health, University of California, Los Angeles, CA 90095, USA; 4David Geffen School of Medicine, University of California, Los Angeles, CA 90095, USA; drhays@ucla.edu; 5RAND, Santa Monica, CA 90401, USA; 6Geriatric Research, Education and Clinical Center, VA Greater Los Angeles Healthcare System, North Hills, CA 91343, USA

**Keywords:** natural disaster, evacuation, transportation, aging, disabilities, chronic disease

## Abstract

Background and Aim: Natural hazard-related disasters such as wildfires, hurricanes, earthquakes, and floods pose significant risks to older adults, individuals with disabilities, and those with chronic health conditions. Transportation-related challenges during and after evacuation can severely impact their safety, mobility, and recovery. This scoping review examines the current evidence to identify research gaps and inform strategies to improve evacuation outcomes and long-term resilience for these at-risk groups. The STEPS framework (Spatial, Temporal, Economic, Physiological, Social) was applied to guide the analysis and interpretation of findings. Methods: This review followed the Preferred Reporting Items for Systematic Reviews and Meta-Analyses Extension for Scoping Reviews (PRISMA-ScR) guidelines and searched five databases, including PubMed, APA PsycINFO, CINAHL Complete, EMBASE, and Web of Science for primary studies on transportation and disaster evacuation among vulnerable populations. Results: Twenty studies were included. Four key areas of concern were identified: (1) immediate transportation barriers during evacuation, (2) prolonged transportation disruptions post-disaster, (3) anticipated logistical challenges in future evacuation planning, and (4) inconsistent and inaccessible communication of transportation-related information during emergencies. These challenges intersected with all five STEPS dimensions. Conclusions: Transportation barriers remain a persistent and under-addressed risk factor in disaster contexts for vulnerable groups. The STEPS framework helped reveal the multidimensional nature of these issues, emphasizing the need for integrated planning, assistive transport options, inclusive communication systems, and stronger public–private coordination. Future research should focus on collecting disaggregated data, evaluating interventions, and supporting inclusive policy reforms tailored to each type of disaster.

## 1. Introduction

Natural hazard-related disasters such as wildfires, hurricanes, earthquakes floods, and tsunamis poses serious and increasing risks to populations worldwide, particularly in the context of global climate change. Among those most vulnerable during such events are older adults, especially those living with disabilities. Research consistently demonstrates that this population faces heightened risks during disaster-related evacuations, with delays, insufficient responses and adverse outcomes disproportionately affecting them [1,2,3,4,5,6,7]. Several factors contribute to this vulnerability, including physical frailty, limited access to resources, and lack of awareness or comprehension of evacuation orders, especially under rapidly changing emergency conditions. These risks are significantly amplified for older adults with disabilities, such as cognitive impairments (e.g., dementia), mobility limitations requiring assistive devices (e.g., canes, wheelchairs), and sensory impairments (e.g., hearing or vision loss), all of which can hinder timely and safe evacuation. For example, during the Eaton wildfire in Los Angeles in January 2025, the average age of those who died in was 77, highlighting the life-threatening consequences faced by this demographic [8]. Similarly, during Hurricane Katrina in 2005, 71% of the fatalities in Louisiana were individuals over the age of 60, and approximately half were 75 years old or older [9,10]. These patterns reflect a broader trend across different types of disasters.

Transportation challenges are a central factor in the evacuation difficulties experienced by older adults and people with disabilities. Public transportation systems are often limited, inaccessible, or not equipped to meet the needs of individuals using mobility aids. Private transportation may not be available or affordable, particularly for those living alone or with limited income. For example, in Los Angeles, California, where disasters like wildfires and earthquakes can disrupt evacuation routes with little notice, these transportation barriers become life-threatening. Similar issues are reported in hurricanes and other large-scale emergencies, where access to safe evacuation options is often restricted [11].

Moreover, reliance on family members or caregivers for evacuation support can lead to dangerous delays, particularly if these individuals are unavailable, unaware of the situation’s urgency, or affected by the disaster themselves. This reliance underscores a critical gap in emergency preparedness, where many older adults with disabilities are unable to timely access evacuation routes or shelters in a timely and safe manner. As a result, they are at significantly greater risk during and after natural hazard-related disasters. Addressing these challenges is essential for promoting resilience and ensuring the safety of vulnerable populations in the face of increasing such disasters. Previous review papers on natural hazard-related disasters have not primarily focused on transportation challenges [3,11,12], thereby limiting the baseline information available to guide policy and community engagement.

To better understand these issues, it is necessary to examine the lived experiences of vulnerable populations during past disasters. Therefore, our research seeks to answer the question: “what are the transportation-related challenges or barriers faced by older adults, individuals with disabilities, and those with chronic diseases during and after evacuations in natural hazard-related disasters?” This scoping review aims to synthesize existing research to identify key transportation-related barriers, highlight gaps in the literature, and explore policy and practice implications that could improve disaster preparedness and evacuation for at-risk populations.

## 2. Methods

We used the PRISMA-ScR search methodology [13]. This search approach was selected for its systematic, comprehensive, and transparent characteristics.

### 2.1. Search Strategy

A systematic literature search was performed across five databases: PubMed, CINAHL, PsycInfo, EMBASE, and Web of Science, targeting peer-reviewed papers. The four sets of keywords “emergency OR transportation,” “older adults OR disabilities,” “natural disasters,” and “experiences” were used in each database. The controlled vocabulary in all databases, excluding Web of Science, and Boolean logic, was employed in the search. Furthermore, synonyms for every keyword were incorporated into the literature search (see Appendix A, Table A1).

### 2.2. Eligibility Criteria and Search Results

The inclusion criteria for this review were as follows: (1) articles written in English, (2) participants included older adults (55 years old and above), adults of any age with disabilities or chronic illnesses, or official/unofficial support personnel for these populations, (3) studies investigating experiences, including challenges or barriers related to evacuation during natural disasters and their impact on post-disaster life, specifically in relation to transportation, and (4) primary research articles. The eligibility included all types of natural disasters. To ensure comprehensive coverage of the limited available evidence on this topic, no publication start date restrictions were applied. However, the search was limited to studies published up to 19 February 2025.

The exclusion criteria included publication types that were not primary research, such as review articles, editorials, and conference abstracts. To manage the high volume of initial search results, filters were applied to restrict results to studies published in English and involving middle-aged and older adult populations. Although the primary population of interest was adults aged 55 and older, individuals aged 45–64 were also included in the filtering process to reflect the aging continuum and provide a more inclusive view of evacuation-related vulnerabilities. The age threshold of 55 was selected based on the earlier onset of chronic conditions in this group and its alignment with definitions used by various health systems, programs, and research studies [14,15,16]. Where possible, additional filters were applied to exclude conference proceedings. Only Embase and Web of Science offered specific functions to remove conference-related documents.

The final literature search yielded 1195 articles from five databases. An additional six articles were identified through manual search, such as reviewing citations from included studies. After removing duplicates, 960 articles remained for initial screening. Of these, 853 were excluded based on title and abstract review. During full-text screening of 107 articles, studies were considered if they addressed transportation-related challenges in the context of natural disasters. Following the full-text review, 87 articles were excluded for the following reasons. Thirty-nine of the articles did not examine transportation-related challenges, 16 were not primary research articles, 11 did not address evacuation experiences, 10 pertained to emergency technology in transportation, 7 lacked participants who were either older adults or adult individuals with disabilities, two articles were result from the same interview with the same participants and the same results, one article was location-specific, and one article was unavailable in full text. A total of 20 studies met all inclusion criteria and were included in the final review. The study selection process is illustrated in the PRISMA flow diagram (Figure 1).

### 2.3. Study Selection and Data Extraction

This study team conducted a two-phase review process collaboratively. In the first phase, three reviewers screened all titles and abstracts based on the inclusion and exclusion criteria to identify potentially relevant studies. In the second phase, two reviewers independently assessed the full texts of the selected articles to determine final eligibility. Any discrepancies were resolved through discussion and clarification among the reviewers. To guide data extraction, the same two reviewers jointly developed a data-charting form. Using an iterative approach, they charted the data, discussed their findings, and continually refined the form to ensure accuracy and consistency (see Appendix B, Table A2). The final synthesis was a collaborative effort by the team. A team member involved in both screening phases initially developed a summary table of the included studies. This table was reviewed by another team member who also participated in the screening process, and any disagreements were resolved through discussion Finally, two additional team members reviewed the summary table to ensure clarity and consistency.

### 2.4. Theoretical Framework

The STEPS (Spatial, Temporal, Economic, Physiological, and Social) transportation equity framework guided our analysis of the selected studies to identify mobility-related challenges faced by older adults, individuals with disabilities, and those with chronic illnesses. Developed by Shaheen et al. (2017), the STEPS framework categorizes transportation barriers into five dimensions, each representing a critical area where inequities may occur [18]. Spatial factors refer to challenges related to distance and location, such as excessively long distances between destinations or the lack of public transit within walking range. Temporal factors involve travel time barriers that prevent timely trips, including unreliable public transit, limited operating hours, and traffic congestion. Economic factors encompass direct costs (e.g., fares, tolls, vehicle ownership) and indirect costs (e.g., smartphone, internet, credit card access) that can create financial hardship or exclude users from traveling. Physiological factors include physical and cognitive limitations that make it difficult or impossible to use standard transportation modes (e.g., infants, older adults, individuals with disability). Social factors involve barriers related to social, racial, cultural, safety, and language issues, such as neighborhood crime or the lack of multilingual information, which can restrict transportation access. This structured approach allowed us to systematically summarize key challenges and identify gaps in the literature. By clearly delineating areas in need of improvement, the framework supports the development of targeted interventions to enhance transportation equity for vulnerable populations. Although originally designed to assess shared mobility services, the STEPS framework has broader applicability and has been effectively used in other contexts, including evacuation planning and emergency response, due to its comprehensive and flexible structure [19].

## 3. Results

### 3.1. Characteristics of Identified Studies

This research evaluated a total of 20 academic journal articles (see Appendix B). The characteristics of the reviewed articles are summarized in Table 1. The studies were predominantly conducted in Western countries. The majority of the research, 13 in all, was conducted in the United States (USA) [5,19,20,21,22,23,24,25,26,27,28,29], including one in Puerto Rico [30]. Two studies were held in Australia [31,32] and two in New Zealand [33,34]. A limited number of studies were performed in other regions, including India [35], Sri Lanka [36], and Turkey [37].

The studied natural disasters included wildfires [5,26,27], hurricanes [21,25,28,29,30], flood [31,35], and earthquakes [33,34,37]. Two investigations did not specify a particular occurrence [32,36]. Three studies [5,26,27] concentrated on wildfire evacuation, all of which were conducted in California, USA.

Most of the publications identified in this search used qualitative inquiry as their study method, employing either focus groups or interviews [19,22,23,25,27,30,31,32,33,34,36,37]. Among the 20 studies, only six were cross-sectional surveys [5,22,24,25,26,28,35].

The participants were categorized into four groups: older adults ages 55 years and older, individuals with disabilities, individuals with chronic illnesses, and stakeholders (e.g., facility staff, healthcare providers) serving these groups. Nine research targeted older adults [21,22,23,25,27,29,31,32,37]. One study involved adults of varying ages, but it contained specific information regarding older adults [35]. However, the age range of the older adult participants in this study was not specified. Four research studies focused on individuals with disabilities [20,33,34,36]. One study focused on individuals with vision impairments [33]. The remainder included various disabilities, including physical, cognitive, and intellectual disabilities [20,34,36]. Three studies included both older people and those with disabilities [5,19,28]. One study included all targeted groups, older adults, adults with disabilities, and those with chronic diseases [24]. Two studies included both individuals in the targeted groups and their caregivers (i.e., family members) [19,36]. One included individuals with disabilities and their caregivers [36]. Another included both older adults, individuals with disabilities, and their family members [19]. Five studies included only participants in supportive positions, such as nursing home staff, healthcare professionals, and family caregivers [21,23,26,27,30]. Three publications examined professional providers or paid caregivers for older adults [21,23,27], and two worked with chronically ill individuals [26,30].

### 3.2. Identified Key Issues

Despite a dearth of existing scholarly articles, the current literature review identified four key transportation-related challenges during and after evacuations from natural hazard-related disasters affecting vulnerable populations, including older adults, individuals with disabilities, and those with chronic illnesses. These challenges include: (1) immediate transportation barriers during evacuation, (2) prolonged transportation disruptions post-disaster, (3) anticipated logistical challenges in future evacuation planning, and (4) inconsistent and inaccessible communication of transportation-related information during emergencies. Each of these challenges is examined with reference to the STEPS framework, highlighting their mutidimensional impacts and guiding potential avenues for intervention.

#### 3.2.1. Immediate Transportation Barriers During Evacuation (STEPS Dimensions: Spatial, Temporal, Economic, Physiological, Social)

This review highlights the immediate impact of transportation-related challenges on evacuation during natural hazard-related disasters [21,22,23,25,28,29,34,35,36]. Transportation challenges significantly influenced decision-making during emergencies [22,28]. Multiple factors shaped these immediate impacts.

Spatial Barriers

A key spatial challenge was the inability of many older adults, individuals with disabilities, and those with chronic illnesses to access transportation during emergency evacuations [21,23,29,35]. For example, 41% of older adults who remained behind during Hurricane Katrina cited lack of transportation as the main reason for not evacuating [29]. Similarly, a study from the 2018 Kerala floods in South India reported that limited seat availability constrained the evacuation of entire families [35]. Even in institutional settings, Transportation access was not guaranteed. Post-Katrina assessments completed by administrators indicated that only 55% of them (*n* = 6 of 11) reported improvement in evacuation transportation, while others continued to experience difficulties arranging transport for residents during Hurricane Gustav in 2008; both disasters occurred in Louisiana, USA [23]. Infrastructure disruptions, such as road closures and damaged public transit systems, further exacerbated spatial inaccessibility [21].

Temporal Barriers

Timing also played a critical role in evacuation decision-making. Anticipation of traffic congestion led many to delay or forgo evacuation altogether. During Hurricane Irma in 2017, older adults were less likely to evacuate compared to younger individuals. More than half of surveyed individuals reported traffic delays as the main deterrent to evacuating [28]. Moreover, even when individuals opted to evacuate, concerns about traffic influenced their evacuation behavior. Individuals with severe concerns regarding traffic departed before the hurricane’s landfall, with some leaving three days in advance or even earlier. They chose to travel at night to avoid excessive congestion and opted for highways or major roads rather than local streets. However, individuals concerned about gasoline availability opted for local roads over highways because they allowed quicker access to gas stations [28].

Physiological Barriers

Physiological needs added further complexity to evacuation logistics. Medically vulnerable individuals, particularly those dependent on life-sustaining equipment such as ventilators, required specially equipped vehicles, which were often in short supply during emergencies [21]. These individuals also needed healthcare personnel to accompany them during evacuation. However, staffing shortage limited the number of medically complex patients who could be transported safely at one time [21,34].

Social Barriers

The social dimension was evident in the lack of qualified personnel available to assist vulnerable individuals during evacuations. Many older adults and individuals with disabilities rely on family members, caregivers, or medical staff for transportation and support. The limited availability of these support systems during emergencies significantly reduced evacuation capacity [21,34]. Furthermore, prior negative experiences with evacuation, particularly those marked by inadequate transportation affected future evacuation decisions. For example, individuals who had faced transportation challenges during Hurricane chose not to evacuate during future natural disasters [22].

#### 3.2.2. Prolonged Transportation Disruptions Post-Disaster

Transportation difficulties persist for survivors long after a natural disaster has passed. For older adults, individuals with disabilities, and those with chronic illnesses, many of whom rely heavily on public transportation, the prolonged disruption of transit systems can create ongoing challenges [20,24,26,27,30,31,37]. These challenges can impact various aspects of daily life, illustrating the complex and significant long-term repercussions of transportation-related barriers.

##### Impact on Daily Life and Independence (STEPS Dimensions: Spatial, Temporal, Economic, Social)

The disruption of public transit systems limits access to essential destinations such as grocery stores, workplaces, healthcare providers, and social services [20,34]. This loss of mobility directly affects individuals’ independence and quality of life, particularly for those who do not drive or own a private vehicle. Social interactions are also reduced, as traveling to visit family or receive visitors becomes more difficult [31,37]. These challenges are further exacerbated when survivors are relocated to unfamiliar areas, where they must navigate new and often inaccessible transportation systems, posing additional spatial and cognitive barriers, especially for individuals with limited mobility or special needs [20].

In many cases, transportation in the post-disaster phase often relies on informal support networks, such as family members, friends, or professional caregivers [20]. However, these caregivers are frequently disaster survivors themselves and may also face logistical barriers or displacement. The social and economic strain intensified when infrastructure remains damaged or when transportation support services are unavailable [20]. For example, after the Christchurch earthquakes in New Zealand, many survivors were forced to use taxis for medical appointments or travel longer distances to access essential services, which resulted in a significant economic burden [34].

##### Impact on Health (STEPS Dimensions: Spatial, Temporal, Social)

A critical concern in the aftermath of natural disasters is maintaining access to healthcare. Spatial and temporal barriers such as damaged roads or unavailable transit options can delay or completely prevent individuals from receiving necessary medical care [20,26,30]. One study reported that over 40% of oncology patients in California experienced transportation-related barriers after a wildfire, disrupting essential treatments like chemotherapy or radiation therapy [26]. Similarly, individuals requiring dialysis faced significant transportation obstacles with delayed or missed treatments leading to deterioration in health and in some cases, fatalities [30].

Transportation disruptions also impact healthcare systems at a broader level. When healthcare workers are unable to reach their workplace, even operational clinics may struggle to meet patient demand, reducing access to care despite physical infrastructure remaining intact [26,27].

Beyond physical health, transportation challenges contribute to negative mental health outcomes [24,31]. The inability to access social networks or services due to transit barriers leads to heightened feelings of loneliness and helplessness among older adults and other vulnerable populations. This social isolation further compounds the psychological toll of disaster recovery.

#### 3.2.3. Anticipated Logistical Challenges in Future Evacuation Planning (STEPS Dimensions: Spatial, Temporal, Economic, Physiological, Social)

Past evacuations have revealed ongoing gaps in preparedness and highlighted critical needs, especially for vulnerable populations such as older adults, individuals with disabilities, and medically fragile patients These lessons point to a range of anticipated barriers and potential facilitators for future evacuations, with implications across all five domains of the STEPS framework.

Spatial and Economic Barriers

A persistent challenge identified across multiple studies is the limited availability of accessible transportation options, particularly in resource-constrained regions [23,25,28,36]. Concerns about future evacuation logistics extended beyond public and institutional transport to include private services such as ride-shares [5,19]. Individuals with disabilities also raised concerns about whether ride-share services would accommodate their service animals. Older adults reported doubts about the availability and reliability of drivers during the evacuations, as well as worries about potentially high costs [19]. As a result, many preferred carpooling with local volunteer networks.

Temporal and Social Considerations

The timing of transportation availability and support coordination was another major theme. Nursing home administrators specifically expressed concern over the decline in residents’ health after returning to the facility, attributing it to the insufficient or even traumatic nature of evacuation experiences [23]. In response to these barriers, social networks emerged as informal facilitators. Households that included older adults were more likely to organize or participate in carpooling efforts within their communities, particularly in the absence of formal transportation options [5].

Physiological Needs and Suggested Interventions

Evacuating individuals with complex health needs requires transportation that can accommodate medical equipment, mobility aids, and trained assistance. Drawing from past experiences, older adults and individuals with disabilities have offered practical suggestions to address transportation barriers [19,25]. For example, some older adults have recommended using large vehicles, such as school buses, to improve evacuation capacity [25]. Others, including individuals with various disabilities, proposed building a partnership with paratransit services to ensure that people with special needs can be identified and supported during evacuations [19]. Additionally, the need for training drivers to provide better assistance during emergencies was strongly emphasized [19,25].

#### 3.2.4. Inconsistent and Inaccessible Communication of Transportation-Related Information During Emergencies (STEPS Dimensions: Spatial, Temporal, Economic, Physiological, Social)

Multiple participants in the reviewed studies emphasized both the importance and the difficulty of accessing accurate, up-to-date transportation information during emergency evacuations [32,33,34,36]. While not a transportation service itself, the effective dissemination of information is crucial to ensuring that available transit options are used efficiently and safely.

Spatial and Temporal Relevance of Information

For many individuals with disabilities who rely on public transit, both the unavailability of transportation and the lack of timely, accurate information have been identified as significant barriers [34,36]. Three studies noted the absence of reliable information delivery systems, which led to confusion and difficulty navigating during evacuations [33,34,36].

Physiological and Sensory Accessibility

Research involving individuals with various disabilities underscores the need for a multimodal information delivery system. For those with visual impairments, written text is ineffective, making the radio a preferred medium for receiving emergency updates [33,36]. Conversely, individuals with hearing impairments found text-based alerts to be highly effective [33]. Those with both hearing and vision impairments were found to benefit from a Global Positioning System (GPS) based system as a critical source of navigation and information [33].

Social and Economic Considerations in Technology Use

The adoption of digital communication tools also intersects with social and economic dimensions. For example, a study examining the use of a smartphone application to disseminate emergency information to older adults during severe weather events reported positive outcomes [32]. Many of the older adults were already familiar with smartphones and used them for tasks such as managing medical appointments and checking transportation schedules. They found the app to be a valuable tool for receiving emergency updates during extreme weather conditions [32]. However, this approach may not reach individuals without access to smartphones or internet connectivity, pointing to potential economic and digital divide-related barriers. Furthermore, individuals who are socially isolated or not digitally literate may be unable to benefit from app-based communication, highlighting the importance of integrating low-tech and community-based communication methods into emergency preparedness strategies.

## 4. Discussion

This scoping review examined transportation-related challenges affecting vulnerable populations, including older adults, individuals with disabilities, and those with chronic illnesses during and after natural disaster-related disasters. The findings, synthesized from 20 studies across diverse global contexts, were analyzed through the lens of the STEPS transportation equity framework. Despite contextual differences in disaster type and geographic setting, several consistent themes emerged, highlighting critical gaps in emergency transportation equity for at-risk populations. These included immediate evacuation challenges, prolonged post-disaster mobility and health issues, anticipated barriers to future evacuations, and limitations in the dissemination of transportation information.

The STEPS framework revealed that transportation barriers are deeply interwoven and often disadvantage vulnerable individuals across multiple dimensions. Spatial challenges such as inaccessible routes, limited transit availability, or relocation to unfamiliar areas remain a persistent obstacle both during evacuation and in long-term recovery. For instance, during Hurricane Katrina and the Kerala floods, lack of spatial access to transportation services was a key reason that individuals were unable to evacuate safely [25,35]. Similarly, post-disaster infrastructure damage severely limited mobility, affecting daily functioning and healthcare access [20,34].

Temporal factors also played a significant role, particularly in shaping evacuation decisions. The fear of traffic congestion or delays led many to either evacuate prematurely, sometimes days before official warnings, or not evacuate at all [28]. Disruptions to the timing and availability of transportation services further exacerbated difficulties for people with chronic health conditions, as delayed access to dialysis, chemotherapy, or staff at healthcare facilities posed life-threatening risks [26,30].

Economic inequities consistently influenced evacuation behavior and post-disaster recovery. Those without private vehicles or resources to pay for taxis, fuel, or ride-shares were often stranded or forced to depend on unreliable informal networks. In several studies, even when ride-share services were available, high costs or concerns about accessibility (e.g., for service animals) deterred use among older adults and individuals with disabilities [19,36]. Economic barriers were also evident in post-disaster contexts, where survivors had to pay out-of-pocket for alternative transport due to ongoing public transit disruptions [34].

Physiological needs, especially among medically fragile individuals further limited evacuation and recovery options. The shortage of accessible vehicles and trained staff capable of supporting medical transport during emergencies was a recurring concern. For patients reliant on ventilators, dialysis, or ongoing treatment, the lack of equipped evacuation services translated directly into increased morbidity and, in some cases, mortality [21,26]. These challenges are not only infrastructural but also policy-related, pointing to the need for proactive integration of healthcare-dependent populations into transportation planning for disasters.

Social factors amplified many of the other STEPS dimensions. Support systems such as caregivers, family members, or healthcare staff were often themselves affected by disasters, creating cascading failures in care and mobility for vulnerable individuals. In addition, social isolation, especially among older adults, contributed to poorer evacuation outcomes and increased psychological distress in the aftermath of disasters [24,31]. Community-based efforts, such as local carpooling networks, showed promise in some studies [5], yet these informal mechanisms cannot substitute for comprehensive institutional support.

Another key theme was the dissemination of transportation information during emergencies. Many participants, particularly individuals with sensory impairments, faced barriers accessing evacuation-related information in a timely and usable format. The mismatch between communication modes and users’ needs (e.g., written notices for the visually impaired or radio alerts for the hearing impaired) exposed significant gaps in inclusive emergency communication planning [33,36]. While mobile apps and GPS-based systems showed promise in enhancing information access [32], their benefits may be limited by digital literacy, internet access, and socioeconomic status.

Taken together, the results emphasize that transportation equity during natural disasters must be approached as a multidimensional problem. The STEPS framework not only captures the intersectional nature of transportation barriers but also provides a comprehensive structure for evaluating current shortcomings and guiding future interventions.

The findings about the transportation challenges encountered by older adults, individuals with disabilities, and those with chronic illnesses during and after natural disasters are consistent with prior literature reviews. However, most previous reviews looked at just one of the vulnerable groups reviewed in this study. Transportation interruptions experienced by older people are known to impede their resilience during disasters [38]. This review documented the enduring issues faced by older individuals following a disaster, resulting in adverse effects on their mental, emotional, and physical well-being [38]. Individuals with disabilities and chronic illnesses faced limitations in transportation, particularly accessible services, after natural hazard-related disasters [11]. The enhancement of public transportation options may yield increased resilience during natural hazard-related disasters and foster greater community involvement for individuals with disabilities and chronic illnesses [11]. The review of hospital systems and their evacuations indicated that transportation is a critical component influencing the success or difficulty of the evacuation process [39]. The availability of pre-existing ambulance resources is a critical factor in successful evacuations. In contrast, transportation issues, including insufficient transport methods and challenges in moving patients with medical needs, constituted significant obstacles during patient evacuations [39]. Mortality rates among older adults and those with impairments increase during natural hazard-related disasters [40,41]. The challenges encountered during evacuation were identified as a potential factor contributing to the increased fatality rate [40]. Enhancing transportation during and after evacuation may reduce the victim count among these at-risk populations.

Experiences with previous natural hazard-related disasters and the challenges faced by older adults, individuals with disabilities, or those with chronic illnesses may influence their future evacuation decisions. This review identified inconsistent results on the role of age on evacuation rate: some studies showed that older adults are more likely to evacuate [1,7], but others suggested that they are more likely to stay in place [22,25]. It is essential to comprehend these evacuation patterns with more precision. GPS data from mobile devices can provide a more precise analysis of evacuation behaviors [7]. This information will facilitate the development of effective disaster plans to support vulnerable people and enhance overall community preparation for natural hazard-related disasters.

### 4.1. Policy Implications

This review identified transportation challenges faced by vulnerable populations during and after natural hazard-related disasters. These concerns have multi-dimensional, negative consequences, including both direct and indirect effects, as well as immediate and long-term impacts on their health and well-being. Inadequate disaster preparedness and limited community resources pose a serious threat to community resilience in the face of natural hazard-related disasters [42]. Applying the STEPS framework, this review identified critical gaps in spatial, temporal, economic, physiological, and social dimensions of evacuation and transportation. The following policy recommendations are directly grounded in specific findings from the review and are aligned with international frameworks such as the Sendai Framework for Disaster Risk Reduction 2015–2030 [43], the UN Madrid International Plan of Action on Ageing [44], and the WHO Emergency and Disaster Risk Management (EDRM) guidelines [45]. The Sendai Framework for Disaster Risk Reduction 2015–2030 emphasizes the importance of enhancing the resilience of new and existing critical infrastructure, including transportation, to ensure that systems remain safe, effective and operational during and after disasters, thereby providing live-saving and essential service. The UN Madrid International Plan of Action on Ageing highlights that older adults are particularly vulnerable in emergencies such as natural hazard-related disasters and should have equitable access to all services during and after such events, although it does not include transportation specific action plans. The WHO EDRM guidelines further point out the importance of maintaining critical infrastructure, transportation, logistics, and emergency services, and access to care for older adults before, during, and after emergencies.

#### 4.1.1. Improve Emergency Transportation Access for Vulnerable Populations

Lack of accessible and reliable transportation was the most frequently cited barrier to evacuation among older adults and individuals with disabilities [23]. Even institutional providers like nursing homes reported difficulty securing transportation post-Katrina [23].

Policy recommendation: Government at all levels should mandate inclusive transportation planning for vulnerable populations. Current guidance, such as California’s 2023 integrated evacuation planning for individuals with access and functional needs [46], delegates planning to local jurisdictions without requiring enforcement or standardization. Jurisdictions should be required to:Develop formal evacuation transportation plans for individuals with medical equipment needs (e.g., ventilators).Partner with paratransit and medical transport providers to ensure vehicle and staff availability.Enforce transportation contracts and evacuation drills for long-term care facilities.Establish public subsidies for ride-share or accessible transportation options for low-income evacuees. These steps would directly address the spatial, physiological, and economic barriers documented in this review.

#### 4.1.2. Mandate Baseline Data Collection and Transportation Mapping

A lack of granular data on transportation needs limits planning efforts, particularly in regions with high disaster exposure and infrastructure vulnerabilities [35,36,37].

Policy recommendation: Authorities should be required to:Conduct needs assessments for at-risk groups using local demographic data.Map evacuation routes and identify chokepoints using GIS tools (e.g., narrow roads, single-exit neighborhoods).Identify populations aging in place or with multiple disabilities who may lack formal caregivers. This supports spatial and economic planning and responds directly to the challenges faced by individuals unable to evacuate due to infrastructure limitations or lack of caregiver support [25,35,36].

#### 4.1.3. Enhance Post-Evacuation Transportation Support

Disruption of public transportation systems post-disaster led to social isolation, missed medical appointments, and long-term decline in independence and mental health [20,26,27,31,37].

Policy recommendations:Include post-disaster transportation recovery into national and regional disaster risk management strategies.Allocate funding for temporary transport services (e.g., shuttle routes to healthcare, pharmacies, or grocery stores).Expand support for displaced populations who no longer have access to familiar transit systems.

#### 4.1.4. Developing Multimodal and Accessible Emergency Communication Systems

Individuals with sensory impairments had unequal access to transportation information during disaster due to poor communication formats [33,36]. Preferences varied: those with visual impairments preferred radio; those with hearing impairments preferred text; those with dual impairments relied on GPS [33].

#### 4.1.5. Train and Mobilize Transportation Personnel for Disaster Response

The review found that during evacuations, a shortage of trained personnel limited the number of medically complex individuals who could be transported safely [21,34]. Transport providers often lacked training to assist passengers with medical equipment or cognitive impairments.

#### 4.1.6. Incentivize Community-Based Transportation Solutions

Informal support systems such as neighborhood carpools proved essential during past evacuations, especially when formal services were insufficient. Households with older adults were more likely to coordinate carpooling [5,19].

Policy recommendation:Fund neighborhood-based transportation networks, including volunteer ride registries.Incentivize local governments and nonprofits to maintain community transportation response plans using local assets (e.g., churches, school buses).Establish legal and insurance protections for volunteer drivers during declared emergencies.

These actions align with the social and economic facilitators identified in the review and capitalize on existing informal networks.

### 4.2. Limitations

While this review highlights critical transportation challenges faced by vulnerable populations during natural hazard disasters, several limitations should be acknowledged. First, most of the studies are qualitative with only approximately 35% employing quantitative methods. While qualitative studies offer valuable insights into individual experiences, their findings may not be generalizable.

Second, because public policy and infrastructure vary significantly across regions, some of the findings and policy implications may not be universally transferrable. This review includes studies from various global contexts, and while certain transportation-related issues are shared across disaster types and locations, other challenges are likely to be highly context- or hazard-specific.

Third, while this review included some studies from non-Western countries, the majority of the literature analyzed originated from Western contexts. As such, the findings may reflect sociopolitical and infrastructural assumptions that are not applicable in lower-resource or culturally distinct settings. Therefore, the generalizability of the results should be interpreted with caution, and further research is needed to explore transportation challenges in underrepresented global regions.

### 4.3. Future Research Directions

Currently, there is a lack of baseline data on the specific transportation needs of older adults and individuals with disabilities during evacuations in natural hazard-related disasters. Future research should address this gap by examining the transportation experiences of those affected by specific natural hazard-related disasters such as recent wildfires in Los Angeles, including individuals in both “red” or immediate evacuation zones and those in “yellow” or pre-evacuation zones. Key areas of inquiry should include the types of transportation options available to them, their effectiveness, and the role of support systems such as rides from family or friends, as well as the use of technology (e.g., real-time wildfire status apps).

There is a need for objective data to identify and assess transportation-related risks in geographically vulnerable areas. This includes identifying high-risk regions with infrastructure issues, such as one-way exits, narrow roads, or limited access points, that could hinder emergency evacuation efforts. Additionally, it involves assessing resources, including accessible transportation services, in each region. Furthermore, it is critical for identifying the locations of especially vulnerable persons. Currently, aging within the community and communal living for those with disabilities have been promoted to enhance independence and quality of life. Nevertheless, as this review indicates, older adults are more susceptible to the effects of natural disasters. Consequently, it is essential to identify individuals in each community who may require transportation assistance, facilitating prompt intervention during natural disasters. Municipalities may employ census data and partner with agencies that provide services for older adults or individuals with disabilities.

As the number of adults with sensory deficits (e.g., hearing or visual impairments) increases as they age, and the population of individuals with disabilities grows in each community, it is imperative to implement disaster planning and intervention strategies that facilitate resilience and successful evacuation behaviors, tailored to diverse support needs and using appropriate methods of information dissemination. Researchers can develop and evaluate the effectiveness of evacuation and transit alternatives for each demography by employing various technologies to deliver relevant information that is responsive to a range of individual disabilities.

In addition, while qualitative research methods such as interviews and focus groups have provided valuable insights into individuals’ lived experiences during and after evacuation, their findings may not reflect the broader population. Therefore, future studies should incorporate alternative or complementary methodologies, such as quantitative surveys developed using a participatory research approach to capture a more inclusive and representative range of perspectives.

Lastly, because this review integrated findings from studies involving various types of natural hazard-related disasters including hurricanes, wildfires, floods, and earthquakes, future research should investigate whether disaster-specific variables affect transportation access and evacuation success. Comparing transportation challenges across disaster types may uncover critical differences that can inform more tailored planning, policy, and resource allocation.

## 5. Conclusions

The increasing frequency of natural disasters due to climate change has disproportionately impacted older adults, people with disabilities, and those with chronic illnesses largely due to inadequate transportation during and after evacuation. Safe, accessible transit is critical not only for evacuation but also for maintaining health and independence in the aftermath. Using the STEPS framework, the findings from this research show that these barriers affect evacuation decisions, access to healthcare, and long-term recovery, and deepen existing vulnerabilities. Gaps in policy implementation, data collection, and inclusive planning persist. To build community resilience, tailored interventions are needed, such as inclusive transportation systems, multimodal communication strategies, and locally coordinated support networks. Future research should focus on involving local communities in the design of studies that collect both qualitative and quantitative data, examine disaster-specific challenges, and evaluate practical solutions to better support vulnerable populations in emergencies.

## Figures and Tables

**Figure 1 ijerph-22-01680-f001:**
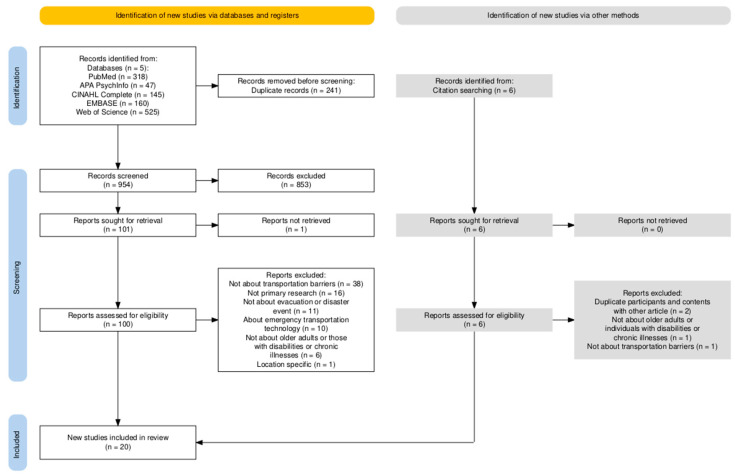
PRISMA 2020 Flow Diagram of the Literature Search on Challenges Faced by Older Adults with Disabilities During and After Evacuations in Natural Disasters [17].

**Table 1 ijerph-22-01680-t001:** Characteristics of Reviewed Studies.

Characteristics	# of Studies	% of Studies	References
Study location			
USA	13	65%	[5,19,20,21,22,23,24,25,26,27,28,29,30]
Australia	2	10%	[31,32]
New Zealand	2	10%	[33,34]
India	1	5%	[35]
Sri Lanka	1	5%	[36]
Turkey	1	5%	[37]
Study methodology			
Qualitative	13	65%	[19,20,21,23,25,27,30,31,32,33,34,36,37]
Quantitative	7	35%	[5,22,24,26,28,29,35]
Data type			
Interview	8	40%	[21,23,30,31,33,34,36,37]
Survey	7	35%	[5,22,24,26,28,29,35]
Focus Group	3	15%	[19,20,32]
Document Review	1	5%	[27]
Open-ended Survey questions	1	5%	[25]
Participant size			
<10	1	5%	[32]
<20	4	20%	[21,31,33,36]
<30	4	20%	[19,23,34,37]
>30	10	50%	[5,20,22,24,25,26,28,29,30,35]
N/A (document review)	1	5%	[27]
Type of vulnerable groups			
Older adults	10	50%	[21,22,23,25,27,29,31,32,35,37]
Individuals with disabilities	4	20%	[20,33,34,36]
Individuals with chronic illnesses	2	10%	[26,30]
Older adults and individuals with disabilities	3	15%	[5,19,28]
Older adults, individuals with disabilities, and those with chronic illnesses	1	5%	[24]
Types of participants			
Self	11	55%	[20,22,24,25,28,29,31,32,33,34,37]
Self and family caregivers	3	15%	[5,19,35]
Formal care providers	5	25%	[21,23,26,27,30]
Self- and formal care providers	1	5%	[36]
Types of events			
Wildfires	3	15%	[5,19,26]
Hurricanes	10	50%	[20,21,22,23,24,25,27,28,29,30]
Earthquakes	3	15%	[33,34,37]
Floods	2	10%	[31,35]
Unspecified	2	10%	[32,36]

## Data Availability

No new data were created or analyzed in this study. Data sharing is not applicable to this article.

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
