# Peer review of "Evacuation and Transportation Barriers Among Vulnerable Populations in Natural Hazard-Related Disasters: A Scoping Review"

_ijerph, 2025, doi:10.3390/ijerph22111680_

Round 1
Reviewer 1 Report
Comments and Suggestions for Authors
This paper presents an important topic by addressing the evacuation of vulnerable populations during disasters. However, it does not appear to offer substantial new insights. The manuscript would benefit from more detailed analysis and deeper discussion.
- Abstract: The background and conclusions presented in the abstract are largely overlapping. Consider clarifying the distinction between the research context and the key findings.
- Research Question: The paper lacks a clearly defined research question. Articulating a specific question would help guide the reader and strengthen the overall structure.
- Inclusion Criteria: The time frame for the included publications is not specified. Please indicate the period covered by the literature review.
- Results and Analysis: Although the paper focuses on wildfire preparedness, the results do not sufficiently reveal patterns, gaps, or diversity in the literature. A more thorough analysis is needed to address the stated objectives.
- Policy Recommendations: The discussion section offers general policy suggestions, but lacks specificity. Please include concrete examples of existing policies or systems currently in place.
- Future Research Directions: While only 10% of the reviewed papers are quantitative, the primary goal of a scoping review is to identify research gaps. I recommend providing suggestions for future studies on transportation and evacuation of vulnerable populations, including variations by disaster type.
Author Response
|
|
Comments from Reviewer 1 |
Author’s Response/Revisions |
|
1 |
Abstract: The background and conclusions presented in the abstract are largely overlapping. Consider clarifying the distinction between the research context and the key findings. |
We clarified the Background and Aim and updated the Conclusions accordingly (lines 18-25, 34-40). We also used the STEPS (Spatial, Temporal, Economic, Physiological, and Social) transportation equity framework and updated the key findings in the Abstract (lines 30-34). |
|
2 |
Research Question: The paper lacks a clearly defined research question. Articulating a specific question would help guide the reader and strengthen the overall structure. |
We have now added the following research question to the Background section: “What are the transportation-related challenges or barriers faced by older adults, individuals with disabilities, and those with chronic diseases during and after evacuation in natural hazard-related disasters?”. (Introduction, lines 83-87). |
|
3 |
Inclusion Criteria: The time frame for the included publications is not specified. Please indicate the period covered by the literature review. |
No publication start date restrictions were applied in order to ensure comprehensive coverage of the limited available evidence. However, the search was limited to studies published up to February 19, 2025. This clarification has been added to the Methods section (Method, 2.2. Eligibility Criteria and Search Results, lines 112-114). |
|
4 |
Results and Analysis: Although the paper focuses on wildfire preparedness, the results do not sufficiently reveal patterns, gaps, or diversity in the literature. A more thorough analysis is needed to address the stated objectives. |
Based on the reviewers’ feedback, we broadened the focus to examine transportation challenges during and after disasters related to natural hazards, rather than limiting the scope to wildfires. Key findings are presented within the STEPS framework, which also guided us in identifying research gaps. Notable patterns included spatial, physiological, and social barriers that affected many older adults, individuals with disabilities, and those with chronic illnesses, particularly in terms of accessing transportation during and after evacuations, as well as limited access to evacuation-related communication. Additional barriers included temporal challenges such as traffic congestions, economic barriers such as the high costs of private ride-share services, and limited access to gasoline stations (Results, 3.2. Identified Key Issues, lines 217-388). |
|
5 |
Policy Recommendations: The discussion section offers general policy suggestions, but lacks specificity. Please include concrete examples of existing policies or systems currently in place.
|
Thank you for this comment. Since our revised focus is a broader scoping review of transportation issues across all types of disasters among vulnerable populations, we believe that general policy implications are more appropriate. We also revised the structure of the Policy Implicationssection to highlight key findings organized according to the STEPS framework (4.1. Policy Implications, lines 478-571). |
|
6 |
Future Research Directions: While only 10% of the reviewed papers are quantitative, the primary goal of a scoping review is to identify research gaps. I recommend providing suggestions for future studies on transportation and evacuation of vulnerable populations, including variations by disaster type. |
We concur with this reviewer’s comment and have added suggestions for future studies addressing transportation and evacuation challenges experienced by these vulnerable groups. In particular, we highlight the need for future research to incorporate alternative or complementary methodologies such as quantitative surveys developed using a participatory research approach to capture a more inclusive and representative range of perspectives. While our scoping review included all types of natural hazard-related disasters, we were unable to examine variations by disaster type because of the limited number of available studies. Therefore, we recommend that further research explore the specific needs associated with different disaster types. These additions are now included in Section 4.3. Future Research Directions (lines 620-631). |

Reviewer 2 Report
Comments and Suggestions for Authors
The manuscript is generally well-structured and addresses a significant gap in disaster preparedness literature. However, several critical issues require substantial revision before the paper can be considered for publication. 1) The title specifically promises "Insights for California Wildfire Preparedness". However, only 3 of the 20 included studies (15%) focused on wildfires, with just one of those specifically addressing California. This represents a significant overstatement of the paper's actual contribution. The authors acknowledge this limitation (lines 109-111), but the title, abstract, and introduction continue to overemphasize California wildfire-specific insights that the evidence base cannot support. The paper would be stronger with a more general title reflecting the broader scope of the included studies. 2) While the paper systematically identifies four key transportation challenges, it lacks critical evaluation of the methodological quality of the included studies. A robust scoping review should assess the strength of evidence rather than merely reporting findings. 3) The paper makes California-specific policy recommendations based largely on studies conducted in diverse contexts (Puerto Rico, Australia, New Zealand, India, etc.) without adequately addressing transferability concerns. The authors should more carefully discuss which findings are likely applicable across disaster types and contexts, and which might be specific to particular disasters or regions. 4) The introduction strongly emphasizes California wildfires (lines 40-67), but much of the results section discusses hurricane evacuation findings without clearly connecting them to wildfire-specific challenges. The disconnect between stated focus and actual content creates confusion for readers expecting California wildfire-specific insights. 5) The discussion of "anticipated barriers to future evacuations" (section 3.2.3) contains some repetitive statements that could be condensed. Several policy recommendations (section 4.2) would be strengthened by linking specific recommendations to particular findings from the review. The paper would benefit from incorporating relevant theoretical frameworks for disaster response and transportation planning to strengthen the conceptual foundation.
Author Response
|
|
Comments from Reviewer 2 |
Author’s Response/Revisions |
|
1 |
The title specifically promises "Insights for California Wildfire Preparedness". However, only 3 of the 20 included studies (15%) focused on wildfires, with just one of those specifically addressing California. This represents a significant overstatement of the paper's actual contribution. The authors acknowledge this limitation (lines 109-111), but the title, abstract, and introduction continue to overemphasize California wildfire-specific insights that the evidence base cannot support. The paper would be stronger with a more general title reflecting the broader scope of the included studies. |
We concur with this comment. The revised title is: “Evacuation and Transportation Barriers Among Vulnerable Populations in Natural Hazard-Related Disasters: A Scoping Review". Accordingly, we revised all sections of the paper, including the title, abstract, introduction, results, conclusion, and policy implications.
|
|
2 |
While the paper systematically identifies four key transportation challenges, it lacks critical evaluation of the methodological quality of the included studies. A robust scoping review should assess the strength of evidence rather than merely reporting findings. |
Thank you for this important comment. We agree that methodological evaluation is critical; however, our ability to conduct a systematic appraisal was constrained by the limited information provided in the included studies. Of the 20 articles, the majority were qualitative and descriptive, and many lacked sufficient methodological detail for standardized evaluation. To address this, we emphasized the main limitations observed across the studies, including a lack of disaster-type specificity and restricted generalizability. We clarified this limitation in the revised manuscript and noted the need for future studies to adopt more rigorous and transparent methodologies in the Discussion section (4.2. Limitations, lines 573-589; 4.3. Future Research Directions, lines 620-631). |
|
3 |
The paper makes California-specific policy recommendations based largely on studies conducted in diverse contexts (Puerto Rico, Australia, New Zealand, India, etc.) without adequately addressing transferability concerns. The authors should more carefully discuss which findings are likely applicable across disaster types and contexts, and which might be specific to particular disasters or regions. |
As noted in our responses to other comments, we revised the policy implications to reflect more general perspectives aligned with international frameworks such as the Sendai Framework for Disaster Risk Reduction 2015-2030, the UN Madrid International Plan of Action on Ageing, and the WHO Emergency and Disaster Risk Management guidelines. This shift was made due to the limited evidence available to support highly specific disaster-specific recommendations. For example, the lack of accessible and reliable transportation emerged as the most frequently cited barrier to evacuation among older adults and individuals with disabilities. Policy implications include: developing formal evacuation transportation plan for individuals with medical equipment needs; partnering with paratransit and medical transport providers to ensure vehicle and staff availability; enforcing transportation contracts and evacuation drills in long-term care facilities; and establishing public subsidies for ride-share or accessible transportation options for low-income evacuees (4.1. Policy Implication, lines 478-571). |
|
4 |
The introduction strongly emphasizes California wildfires (lines 40-67), but much of the results section discusses hurricane evacuation findings without clearly connecting them to wildfire-specific challenges. The disconnect between stated focus and actual content creates confusion for readers expecting California wildfire-specific insights.
|
We addressed this comment by broadening our scoping review to include all types of natural hazard-related disasters (Introduction, lines 45-50, 61, 77-90), |
|
5 |
The discussion of "anticipated barriers to future evacuations" (section 3.2.3) contains some repetitive statements that could be condensed. |
We revised the section title to “3.2.3. Anticipated Logistical Challenges in Future Evacuation Planning” and updated the results, accordingly, organizing them within he STEPS dimensions and removing repetitive statements (lines 304-337). |
|
6 |
Several policy recommendations (section 4.2) would be strengthened by linking specific recommendations to particular findings from the review. |
We reorganized the Policy Implications section to highlight key findings structured around the STEPS framework. For example, one of our key findings was that the disruption of public transportation systems post-disaster led to social isolation, missed medical appointments, and long-term decline in independence and mental health. Policy recommendations include: integrating post-disaster transportation recovery into national and regional disaster risk management strategies; allocating funding for temporary transport services; and expanding support for displace populations who no longer have access to familiar transit systems (4.1. Policy Implication, lines 478-571). |
|
7 |
The paper would benefit from incorporating relevant theoretical frameworks for disaster response and transportation planning to strengthen the conceptual foundation. |
As previously responded, this revised version utilizes the STEPS framework (2.4. Theoretical Framework, lines 158-181). |
Reviewer 3 Report
Comments and Suggestions for Authors
The manuscript addresses a relevant and timely topic, focusing on the transportation challenges faced by older adults, people with disabilities, and those with chronic illnesses during and after evacuations in the context of natural hazard–related disasters. However, there are areas that require strengthening in order to consolidate theoretical robustness, integrate relevant antecedents, and develop a more articulated discussion with previous work.
Title: The title is appropriate and clearly reflects the study’s focus. Nevertheless, it could benefit from a minor reformulation to emphasize the broader scope of the review (not restricted solely to wildfires in the California context), considering that the results encompass various types of disasters.
Introduction: While the references employed are pertinent, the introduction lacks integration with the broader literature on aging and disasters. Previous systematic reviews and studies should be explicitly incorporated into the introduction to situate the findings within the consolidated knowledge base. This would help to avoid the impression that the manuscript emerges from an absolute void (which does apply in the specific field of transportation, but not in the wider domain of older adults and disasters).
Relevant works to consider include:
- Sandoval-Díaz, J., Navarrete-Valladares, C., Vega-Ortega, J., Suazo-Muñoz, C., Riquelme, J. P. G., Sandoval-Obando, E., & Valenzuela, C. R. (2025). Community resilience to wildfires: A systematic review of impacts, coping strategies, indicators, and governance challenges. Progress in Disaster Science, 27, 100447. https://www.sciencedirect.com/science/article/pii/S2590061725000444
PRISMA Methodology: Compliance with the PRISMA 2020 checklist is adequate, and the appendices provide transparency and traceability of the review process.
Results and Discussion: The findings are clearly organized, though their presentation could be strengthened through the use of visual schemes or figures to better synthesize the results.
In the discussion, it would be valuable to expand comparisons with previous reviews and provide a more critical analysis of the methodological limitations of the field. For instance, highlighting the scarcity of studies in non-Western contexts, the overrepresentation of qualitative designs, and the implications of these trends—particularly when comparing wildfire-related disasters with other hazards (e.g., hurricanes, floods, earthquakes).
In addition, it is recommended to incorporate the concept of community resilience as it relates to human responses to wildfires, since this approach provides insights into how communities mobilize collective capacities, cope with uncertainty, and develop adaptive strategies when facing large-scale events. This perspective is particularly relevant for linking transportation-related findings to broader processes of social organization and territorial memory in disaster contexts. Key references to be integrated here include:
- Sandoval-Díaz, J., Navarrete-Valladares, C., Vega-Ortega, J., Suazo-Muñoz, C., Riquelme, J. P. G., Sandoval-Obando, E., & Valenzuela, C. R. (2025). Community resilience to wildfires: A systematic review of impacts, coping strategies, indicators, and governance challenges. Progress in Disaster Science, 27, 100447. https://www.sciencedirect.com/science/article/pii/S2590061725000444
Contributions and Policy Implications: The section on policy implications is a strength of the manuscript. However, it could be enhanced by establishing a closer connection with international normative frameworks such as the Sendai Framework, the WHO guidelines, and UN-Aging initiatives, as well as comparative experiences from other countries. Furthermore, it is strongly recommended to use the terminology of socio-natural disasters or natural hazard–related disasters instead of “natural disasters,” in order to align with contemporary approaches in disaster risk management.
Author Response
|
|
Comments from Reviewer 3 |
Author’s Response/Revisions |
|
1 |
Title: The title is appropriate and clearly reflects the study’s focus. Nevertheless, it could benefit from a minor reformulation to emphasize the broader scope of the review (not restricted solely to wildfires in the California context), considering that the results encompass various types of disasters. |
We concur with this comment. The revised title is: “Evacuation and Transportation Barriers Among Vulnerable Populations in Natural Hazard-Related Disasters: A Scoping Review". Accordingly, we revised all sections of the paper, including the title, abstract, introduction, results, conclusion, and policy implications. |
|
2 |
Introduction: While the references employed are pertinent, the introduction lacks integration with the broader literature on aging and disasters. Previous systematic reviews and studies should be explicitly incorporated into the introduction to situate the findings within the consolidated knowledge base. This would help to avoid the impression that the manuscript emerges from an absolute void (which does apply in the specific field of transportation, but not in the wider domain of older adults and disasters). Relevant works to consider include: · Sandoval-Díaz, J., Navarrete-Valladares, C., Vega-Ortega, J., Suazo-Muñoz, C., Riquelme, J. P. G., Sandoval-Obando, E., & Valenzuela, C. R. (2025). Community resilience to wildfires: A systematic review of impacts, coping strategies, indicators, and governance challenges. Progress in Disaster Science, 27, 100447. https://www.sciencedirect.com/science/article/pii/S2590061725000444 |
As we mentioned earlier, we revised the Introduction based on the comments from all reviewers. We also included this reviewer’s suggested article in the Introduction section (lines 77-82). |
|
3 |
Results and Discussion: The findings are clearly organized, though their presentation could be strengthened through the use of visual schemes or figures to better synthesize the results. |
Thank you for this suggestion. While we agree that a visual scheme or figure could be useful for presenting the synthesized results, we believe that a table organized by the STEPS framework is sufficient, particularly given the limited number of articles reviewed (Appendix B, Table 3). Key Findings Aligned with the STEPS Transportation Equity Framework (Spatial, Temporal, Economic, Physiological, and Social). |
|
4 |
In the discussion, it would be valuable to expand comparisons with previous reviews and provide a more critical analysis of the methodological limitations of the field. For instance, highlighting the scarcity of studies in non-Western contexts, the overrepresentation of qualitative designs, and the implications of these trends—particularly when comparing wildfire-related disasters with other hazards (e.g., hurricanes, floods, earthquakes). |
We have now addressed that most studies are qualitative, limiting broader applicability. Regional differences in policy and infrastructure may affect the transferability of findings. Additionally, the predominance of studies from Western context raises concerns about the relevance of conclusions for lower- resource or culturally different settings. Future research is needed in underrepresented global regions. Given the limited number of studies on our topic, this scoping review did not focus on detailed methodological critique. Instead, the aim was to amp existing knowledge, highlight limitations, identify directions for future research, and consider policy implications (4.2. Limitation section, lines 573-589). |
|
5 |
In addition, it is recommended to incorporate the concept of community resilience as it relates to human responses to wildfires, since this approach provides insights into how communities mobilize collective capacities, cope with uncertainty, and develop adaptive strategies when facing large-scale events. This perspective is particularly relevant for linking transportation-related findings to broader processes of social organization and territorial memory in disaster contexts. Key references to be integrated here include: · Sandoval-Díaz, J., Navarrete-Valladares, C., Vega-Ortega, J., Suazo-Muñoz, C., Riquelme, J. P. G., Sandoval-Obando, E., & Valenzuela, C. R. (2025). Community resilience to wildfires: A systematic review of impacts, coping strategies, indicators, and governance challenges. Progress in Disaster Science, 27, 100447. https://www.sciencedirect.com/science/article/pii/S2590061725000444 |
We now highlight the importance of community resilience and incorporate the suggested article into the Policy Implicationssection (lines 478-571). |
|
6 |
Contributions and Policy Implications: The section on policy implications is a strength of the manuscript. However, it could be enhanced by establishing a closer connection with international normative frameworks such as the Sendai Framework, the WHO guidelines, and UN-Aging initiatives, as well as comparative experiences from other countries. |
Because our review focuses on a broader perspective of all types of natural hazard-related disasters, we were unable to include comparisons from other countries. Instead, we have incorporated global frameworks, as suggested by the reviewer (lines 478-501).
|
|
7 |
Furthermore, it is strongly recommended to use the terminology of socio-natural disasters or natural hazard–related disasters instead of “natural disasters,” in order to align with contemporary approaches in disaster risk management. |
We updated the term, changing “natural disasters” to “natural hazard-related disasters.”
|
Round 2
Reviewer 3 Report
Comments and Suggestions for Authors
I consider that this second revision adequately and comprehensively addresses the comments made in the first round of review. The authors have properly incorporated the suggested changes, including the reformulation of the title, the integration of recent and relevant literature, the inclusion of the community resilience framework, the update of terminology, and the connection with international normative frameworks. Overall, the revisions strengthen the manuscript’s coherence, analytical depth, and relevance.
Author Response
We sincerely thank the reviewers and the editor for their additional feedback. In response, we have removed citation #13 from two sections (lines 83-85 and lines 499-500 in the revised version 1) and updated the reference order accordingly in both the text and tables. Since this was a minor revision with no new additions, we incorporated the changes directly into a clean version. We believe the revised version fully addresses all reviewer comments.